# Total Utilization-Upcycling of Mushroom Protein By-Product: Characterization and Assessment as an Alternative Batter Ingredient for Fried Shrimp

**DOI:** 10.3390/foods12040763

**Published:** 2023-02-09

**Authors:** Diego Garcia, Seung Woon You, Ricardo S. Aleman, Joan M. King, Slavko Komarnytsky, Roberta Targino Hoskin, Marvin Moncada

**Affiliations:** 1Department of Food Bioprocessing and Nutrition Sciences, Plants for Human Health Institute, North Carolina State University, Kannapolis, NC 27599, USA; 2School of Nutrition and Food Sciences, Agricultural Center, Louisiana State University, Baton Rouge, LA 70803, USA

**Keywords:** repurposing, waste, food ingredients, sustainability, value-added

## Abstract

Mushroom by-products are economical and eco-friendly raw materials with bioactive and functional characteristics that allow for potential uses as food ingredients. However, mushroom upcycling has yet to be fully exploited, despite the many opportunities that mushrooms may offer. The mushroom protein by-product (MPBP) resulting from mushroom protein production was characterized (chemical composition, physicochemical attributes, and functional properties) and incorporated into plant-based batter formulations to prepare four experimental groups with different ratios (*w/w*, %) of wheat flour (W) to MPBP (100 W, 75 W/25 MPBP, 25 W/75 MPBP, and 100 MPBP). Subsequently, the batter was used for frying batter-coated shrimp, which was evaluated for cooking loss, coating pick-up, oil absorption, and color parameters (*L**, *a**, and *b**). MPBP showed high content of dietary fiber, mainly insoluble fiber (49%), and it is potentially suited for the formulation of high-fiber food products. The MPBP physicochemical attributes pH (11.69), water activity (0.34), *L** (58.56), *a** (5.61), *b** (18.03), and particle size distribution (250–500 µm (22.12%), 125–250 µm (41.18%), 63–125 µm (37.53%), and < 63 µm (0.82%) were noted. Concerning the MPBP functional characteristics, solubility (12.7%), emulsifying activity index (7.6 m^2^/gr), emulsion stability index (52.4 min), water holding capacity (4.9%), and oil holding capacity (4.8%) were reported. Adding MPBP into batter formulations for batter-coated shrimp resulted in higher values of cooking loss, oil absorption, coating pick-up, and *a** color, while lowering *L** and *b** values. The best experimental results were reported for group 75 W/25 MPBP, which indicates that MPBP can potentially be accepted as a novel batter ingredient for partial substitution of wheat flour.

## 1. Introduction

Mushrooms are popular and versatile food ingredients that have long been used for culinary preparations and medicinal purposes. Recently, an increased interest has been shown in inedible mushrooms as a food ingredient for technological and functional applications, especially in the high-protein and clean-label markets [1]. In addition, their high protein content (20–30% in dry matter) and desirable amino acid profiles, mushrooms are also a good source of dietary fiber, mainly chitin and glucans [2]. Chitin is an insoluble fiber with potential prebiotic properties for the gut microbiota [3]. Oyster mushrooms (*Pleurotus ostreatus*) are a popular variety with a low percentage of fat, comprised chiefly of unsaturated fatty acids [4], low calories, and rich in protein, zinc, and vitamins B, C, and D [5]. Furthermore, mushrooms are a natural source of umami substances such as amino acids (such as glutamic and aspartic acids), 5′-nucleotides, and small-molecule peptides, which have been reported as an essential flavor enhancer [6].

For all these reasons, mushroom derivatives have become a popular meat and seafood substitute for consumers seeking healthier diet plans, mainly vegetarians and vegans willing to find affordable, meat-free, good-tasting ingredients [2]. The vegetarian market has grown in the last decade due to an increasing number of consumers following alternative and more sustainable food regimens [1]. This trend has opened new opportunities for using creative and innovative ingredients using food by-products. For example, developing protein concentrates and isolates from new, highly nutritional, sustainable sources and mushrooms has raised much commercial interest. During the process, by-products are generated due to the separation protocol necessary to concentrate mushroom protein. These mushroom protein by-products retain some of the sensory attributes of the original mushrooms, which turn them into potentially acceptable resourceful raw materials for multiple applications [1]. However, mushroom upcycling has not been fully exploited yet, despite the many opportunities that mushrooms may offer.

Frying, a cooking technique with cultural and economic importance, is widely used in both the food service and manufacturing industries [7]. Fried and battered foods are popular in the market due to their well-accepted taste and easy frozen storage. The most common batter ingredient in fried foods is wheat flour. However, there is much interest in substituting wheat flour for cheaper and more nutritive alternatives [8]. Without compromising the taste and texture of the final product. In this study, we assess upcycling alternatives for mushroom by-products and evaluate their use as a food ingredient.

Specifically, we used the by-product of oyster mushroom protein production and evaluated its application as a model batter ingredient for batter-coated shrimp. Initially, we characterized the mushroom protein by-product (MPBP) regarding its nutritional composition, physicochemical parameters (pH, water activity, and color), and technological attributes (emulsifying properties, water and oil holding capacities, and solubility). The MPBP was then used in batter formulations, and its performance was assessed. We hypothesized that the typical umami flavor and the composition of oyster mushrooms enable the successful use of MPBP to partially substitute wheat flour in batter formulations for fried seafood products. Our results unveiled the potential of mushroom protein by-products as value-added ingredients for food preparations and established a roadmap for future applications of mushroom secondary waste streams.

## 2. Materials and Methods

### 2.1. Materials

Dried oyster mushrooms (Mushroom House, Westbury, NY, USA), wheat flour (Organic All Purpose Flour, Arrowhead Mills, Hereford, TX, USA), vegetable oil (Happy Belly, Shreveport, LA, USA), bread crumbs (wheat flour, sugar, spices, and salt; Pereg Natural Foods, North York, ON, Canada), carboxymethyl cellulose powder (CMC, Modernist Pantry, Eliot, Maine, USA), and salt (Food Lion, Salisbury, NC, USA) were used in this study. The batter-coated shrimp was purchased from Beleaf, City of Industry, CA, USA.

### 2.2. Obtaining Mushroom Protein By-Product (MPBP)

Initially, dried oyster mushrooms were ground at 32,000 rpm for 2 min using a high-speed multifunction grinder (HC-2000, Cgoldenwall, Mountain View, CA, USA) to obtain mushroom powder (MP). The MP was used for protein production, and the processing by-product produced MPBP. MP was mixed with water to prepare a solution of 15% (*w/v*). The pH of the mushroom solution was adjusted to 12 with 2 M NaOH and ultrasonicated using an ultrasonic generator (FS-900N, Cgoldenwall, USA) with a 13 mm probe system. The pulse amplitude was adjusted to 50%, and the ultrasound treatment was applied at 4 s/20 min with the obtained solution kept in an ice bath. When producing the MPBP, the pH of the solid pellet was adjusted to pH 7 with white vinegar (Member’s Mark, Bentonville, AR, USA, 1.5% *v/v*) using a ratio of 70:30 (pellet: vinegar), followed by centrifugation at 4050 rpm for 15 min. The obtained solution was then centrifuged at 4050 rpm for 1 h at 4 °C. The supernatant was used for protein production experiments (parallel experiments; data not shown), and the solid by-product pellet was utilized for this study. The supernatant was discarded, and the solid pellet was dried overnight in a vacuum oven (AT19, Across International, USA) at 100 °C under 750 mmHg. The dry residue was ground (Mockmill 200, Mock, Melville, NY, USA) and stored at 24 °C in hermetically sealed containers for further use [9]. 

### 2.3. Characterization of MPBP (Mushroom Protein By-Product)

#### 2.3.1. Chemical Composition

The dietary fiber content was determined according to the AOAC method (991.43, 1995) [10]. The moisture content was determined using a moisture analyzer (HE53, Mettler Toledo, Columbus, OH, USA), while the ash content was determined gravimetrically by incineration at 500 °C (muffle DR200, Yamato, Japan). Lipids were measured in a rapid NMR fat analyzer (Oracle CEM, Charlotte, NC, USA), and the protein content was analyzed by a rapid protein analyzer (Sprint CEM, Charlotte, NC, USA). The difference in the sum of moisture, protein, lipid, and ash was estimated as the total carbohydrate content. All the analyses were performed in triplicate. 

#### 2.3.2. Physicochemical Properties 

A calibrated pH meter (Orion Star A211, Thermo Fisher Scientific, Waltham, MA, USA) was used to measure the pH at room temperature. The pH was determined by homogenizing MPBP samples (1 g) with 20 mL of distilled water for 1 min. The water activity (Aw) was determined using an Aqualab water activity meter (4TE, Meter, Palo Alto, CA, USA). The color was analyzed by using a colorimeter (Chroma Meter CR-5, Konica Minolta, Tokyo, Japan), estimating lightness (*L**), *a** (greenness to redness), and *b** (blueness to yellowness). The particle size distribution of dried MPBP was determined using a sieve shaker (H4325, Humboldt, Elgin, IL, USA) and sieves from 10 mesh (1200 µM) to 120 mesh (125 µM). Samples (0.5 g) were mixed with 50 mL of distilled water, vortexed at 4000 rpm for 5 min, and centrifuged at 4000 rpm for 5 min. An aliquot (25 mL) of the supernatant was transferred to aluminum dishes (f 57 mm) and dried in a vacuum oven at 100 °C for 24 h. Solubility was calculated as in Equation (1).
Solubility (%) = (weight of supernatant (g))/(weight of sample (g)) × 100. (1)

The emulsifying properties of MPBP were determined according to the Zhang et al. (2014) [11] protocol with some slight modifications. Initially, an MPBP emulsion was prepared by mixing 15 mL of 1% MPBP solution (*w/v*) and 5 mL of vegetable oil using a high-shear homogenizer (HQ-2509, Mxbaoheng, Mountain View, CA, USA) at 20,000 rpm for 1 min. 50 µL of the obtained emulsion was transferred to a tube containing 4.9 mL of a 0.1% sodium dodecyl sulfate (SDS) solution. Aliquots were collected immediately after homogenization (T0) and after 10 min (T10). The absorbance of both aliquots was measured at 500 nm (Epoch 2, BioTek, Shoreline, WA, USA) using a 96-well microplate. The emulsifying activity index (EAI) and emulsion stability index (ESI) were calculated as in Equations (2) and (3), respectively:EAI = (2 × 2.303 × A_0_)/(0.25) (2)
ESI = (A_0_ × ∆T)/∆A (3)
where Sw is the weight of the sample (g), A0 is the absorbance at 0 min, and A10 is the absorbance at 10 min, ΔT = 10 min, and ΔA = A_0_ − A_10_. 

The water-holding (WHC) and oil-holding (OHC) capacities were determined according to Tontul et al. (2018) [12]. 1 g of sample was suspended in 20 mL of distilled water and oil (1:20). Samples were centrifuged in a 50 mL tube at 3500 rpm for 10 min. The supernatant was discarded, and the WHC and OHC were gravimetrically determined as Equations (4) and (5), respectively:WHC = (Wtw − Wti)/Ws (4)
OHC = (Wto − Wti)/Ws(5)
where Wti is the weight of the centrifuge tube (g), and Ws is the initial weight of the sample (g). Wtw is the weight of the centrifuge tube without the water supernatant (g), and Wto is the initial weight of the tube without the oil supernatant (g). 

### 2.4. Battered Plant-Based Fried Shrimp Production 

#### 2.4.1. Batter Preparation

The batter and fried-batter coated shrimp preparation is illustrated in Figure 1. Four different experimental groups were prepared using distinct ratios of wheat flour/MPBP: 100 g of wheat flour (100 W), 75 g of wheat flour, 25 g of MPBP (75 W/25 MPBP) (Table 1), 50 g of wheat flour and 50 g of MPBP (50W50MPBP), 25 g of wheat flour and 100 g of MPBP (25 W/75 MPBP), and 100 g of MPBP (100 MPBP) to assess the performance of MPBP as a batter ingredient. Salt (NaCl) and CMC were added at 2.5 and 1.5 g/100 g, respectively. The batter was prepared as described by Rahimi and Ngadi (2014) [8], with some modifications. Initially, distilled water was added to the solid mix (ratios: 1.3:1 for 100 W, 1.5:1 for 75 W/25 MPBP, 2:1 for 50W50MPBP, 2.3:1 for 25 W/75 MPBP, and 2.5:1 for 100 MPBP) and thoroughly mixed for 1 min to warrant complete homogenization. Different ratios of water were used to obtain homogenous solutions for each treatment. 

#### 2.4.2. Preparation of Fried Batter-Coated Shrimp

Samples were deep fried in a compact deep fryer (CDF-100 Series, Cuisinart, Stamford, CT, USA) filled with preheated 1.5 L of soybean oil (Great value, Leander, TX, USA) and maintained at 170 °C, monitored by a digital thermometer (TP101, Vilgen, Saint Albans, UK). Before frying, batter-coated shrimp were battered with four different formulations (Table 1) and breaded using only breadcrumbs. All samples were fried for 4 min and left to drain at room temperature for 3 min after frying.

### 2.5. Evaluation of the Fried Product

Cooking loss (CL) was calculated according to a modified procedure [13]. The plant-based battered and breaded shrimps were weighed before and after frying, and the percentage cooking loss was calculated as Equation (6):CL (%) = (m1 − m2)/m1 × 100 (6)
m1 and m2 are batter-coated shrimp weights (g) before and after frying, respectively.

For batter coating, shrimp were weighed before and after coating. The percentage of coating pick-up (CP) was calculated as Equation (7):CP (%) = (m1 − m2/m2) × 100 (7)
where CL is the cooking loss, CP is cooking pick up, and m1 and m2 are the weights (g) before and after coating, respectively (Adrah et al., 2022) [14].

The fat content of the batter coated shrimp was evaluated before and after frying using the Rapid NMR Fat Analyzer (Oracle CEM, Charlotte, NC, USA). The oil absorption capacity (OAC) was determined using Equation (8). F1 and F2 are the fat contents (%) after and before frying, respectively.
OAC (%) = (F1 − F2)/F1 × 100 (8)

### 2.6. Statistical Analysis

All analyses were performed in triplicate. Statistical data analysis was performed using one-way analysis of variance (ANOVA) followed by the Tukey test (*p* < 0.05) using GraphPad Prism software version 9 (GraphPad Software Inc., San Diego, CA, USA).

## 3. Results and Discussion

### 3.1. Characterization of the Mushroom Protein By-Product

Mushrooms are known for their polysaccharides [4]. Carbohydrates are the primary nutritional compound in MPBP (approximately 78%, Table 2). Remarkably, more than half of the carbohydrate fraction (56%) comprises dietary fiber, mainly insoluble fiber (49%). Similar results were reported [5] when characterizing dry edible mushrooms, *Pleurotus ostreatus*. Mushroom polysaccharides have been used for multiple applications in the food industry. Trehalose, mannitol, β-Glucan, chitin, and chitosan typically appear in mushrooms in considerable quantities [2]. These polysaccharides are commonly used as cryoprotectants, prebiotics, sweeteners, and in food systems [1]. Therefore, MPBP might be an ingredient in the production of food products with high fiber requirements. Mushrooms’ protein and ash levels may vary depending on the species. MPBP has significant residual protein (10%) and ash (8.91%) (Table 2). Due to all these characteristics, mushroom powders and by-products can be used as meat replacers, fat replacers, flour replacers, and salt replacers [15]. *Pleurotus* spp. (*eryngii*, *ostreatus*, and *sajor-caju*) may have a broad range of proteins (3.87–41.6 g) [1].

The particle size of MPBP (Table 3) complies with the requirements for flour products [16]. To a certain extent, the particle size of flour-like products like MPBP can be tailored by selecting appropriate grinding conditions (equipment, intensity, and grinding time) to reach a desired particle size distribution.

Usually, MPBP powders have a brownish color, as confirmed by MPBP’s instrumental color coordinates (Table 3). The particle size is a significant parameter since it affects the viscosity, water, and oil holding capacity of the batter as well as the mouthfeel of the final product [17]. The water activity results registered for MPBP are within a microbiologically safe range since water activity levels of 0.85 or less will inhibit the growth of bacteria, yeast, and mold, making the product shelf-stable [18]. Even though the adjusted pH of the MPBP was 11.69 (a consequence of the protein extraction protocol), acetic acid was used to lower the pH to 7 to make it more palatable. Food ingredients are designed to be used in different formulations with specific functionality.

In this study, we investigated the emulsifying ability of MPBP by measuring both EAI and ESI, essential indexes of the emulsifying ability and the ability to stabilize emulsions, respectively. EAI expresses the amount of oil that can be emulsified per sample unit, while ESI is the rate of phase separation over time and predicts emulsion storage stability [19]. Compared to wheat (EAI = 10.28 m^2^/g and ESI = 130 min [9]), MPBP has both lower EAI and ESI (Table 4). The water-holding capacity (WHC) and oil-holding capacity (OHC) are the amounts of water and oil that can be absorbed per gram of flour, respectively [20]. The WHC, OHC, and solubility are commonly inversely correlated with the particle size [21], and the carbohydrate content affects both the OHC and WHC as if the carbohydrate content increases the OHC and WHC [22]. It has been reported that ingredients with high WHC and OHC are desired for fried food formulation because they can improve the flavor and texture of the final product [23].

### 3.2. Assessment of Fried Batter-Coated Shrimp

Frying is widely used in the food industry and service businesses due to the desirable and unique flavor, crispiness, aroma, and taste of fried foods [24]. Fried battered foods are prepared by dipping a food product in batter with subsequent frying. Batter coating provides a physical barrier that prevents excessive moisture loss and preserves food juiciness while adding crunchy texture, pleasant flavor, and an attractive appearance to fried products [25]. Different ingredients have been tested to enhance batter quality, mainly to improve the uniformity and thickness of the coating, adhesion to food, and general organoleptic attributes [26]. For the food industry, lower cooking loss percentages reflect higher process efficiency. The results (Figure 2) showed that the MPBP addition increased the cooking loss. Higher cooking losses indicated more moisture losses during frying, which affected the product’s sensory characteristics [13]. Lower oil absorption is desirable because excessive oil absorption can negatively affect the fried product’s texture, flavor, crust formation, and nutrition properties [27]. The loss of heat and mass (moisture) transfer can be due to fat content. In our study, the amount of insoluble fiber present in MPBP could increase the capacity of oil absorption in the batter. Similarly, cooking loss, oil absorption, and the percentage of MPBP in the batter were linearly correlated (Figure 3). For both parameters, only 75 W/25 MPBP behaved similarly to wheat (100 W; *p* > 0.05) and would be the treatment of choice for a possible partial substitution of wheat flour in the batter.

The coating pick-up is important because, when properly done, it leads to a thicker and more uniform crust and improves the overall appearance and crispiness of the final product [8]. Interestingly, treatment 100 M had the highest value (*p* < 0.05), while all other groups were similar among themselves (Figure 4).

The frying process and the batter formulation influence the final color of a fried food product. In this study, we standardized the frying process to evaluate the batter formula’s influence on the color of fried batter coated shrimp. It is well-known that the color of fried products affects customer satisfaction and marketability [13]. The result of color analysis on the scale *L* a* b** is presented in Figure 5. Treatments with a higher percentage of MPBP had lower *L** values, which might be attributed to the darker color of MPBP. In addition, it might indicate a lower thermal resistance of MPBP during drying or MPBP protein denaturation [28]. The results indicated that treatment 75 W/25 MPBP would be the best candidate for a possible partial wheat substitution. Sensory analyses will follow to clarify consumer acceptance and organoleptic attributes of fried shrimp with MPBP as a batter ingredient.

Other studies reported that coating pick-up is directly proportional to batter viscosity [29,30]. Fibers have been shown to increase batter viscosity [31], and a significant amount of fiber (56.6%) was detected in MPBP. The viscosity increase can be related to the fibers’ water-holding capacity [32]. Similarly, the oil absorption capacity increased when increasing amounts of MPBP were added to batter-coated shrimp. This phenomenon is compatible with other studies showing an increase in oil absorption when increasing insoluble fiber content, which was considered present in MPBP (49.30%) [33]. Likewise, the higher cooking loss reported for batter coated shrimp with higher amounts of MPBP can be related to the high content of insoluble fiber in MPBP. Finally, the high *a** color and low *L** and *b** values on the MPBP shrimps could be due to Maillard reactions enhanced by the proteins and carbohydrates [34,35] found in MPBP.

## 4. Conclusions

The addition of a mushroom protein by-product to batter formulations for batter coated-shrimp was investigated. The MPBP showed high amounts of dietary fiber, protein, and low in fat. The MPBP physicochemical and functional properties were reported as follows: insoluble fiber (49%), pH (11.69), water activity (0.34), *L** (58.56), *a** (5.61), *b** (18.03), solubility (12.7%), emulsifying activity index (7.6 m^2^/gr), emulsion stability index (52.4 min), water holding capacity (4.9%), oil holding capacity (4.8%), and particle size distribution (250–500 µm (22.12%), 125–250 µm (41.18%), 63–125 µm (37.53%), and < 63 µm (0.82%). For future studies, sensory examination, conjoint analysis, and preference mapping techniques regarding eco-friendly customers should be addressed to understand consumer behavior. Batters prepared with a higher concentration of MPBP led to fried batter-coated shrimp with increased cooking loss, oil absorption, coating, and *a** color, whereas lower values in *L** and *b** values. Therefore, the batter formulation of 75 W/25 MPBP (wheat flour/MPBP ratio) was considered the best choice for a partial substitution of wheat flour due to the most favorable results regarding key attributes of fried batter-coated shrimp. Overall, MPBP proved to be a promising novel ingredient for batter formulations.

## Figures and Tables

**Figure 1 foods-12-00763-f001:**
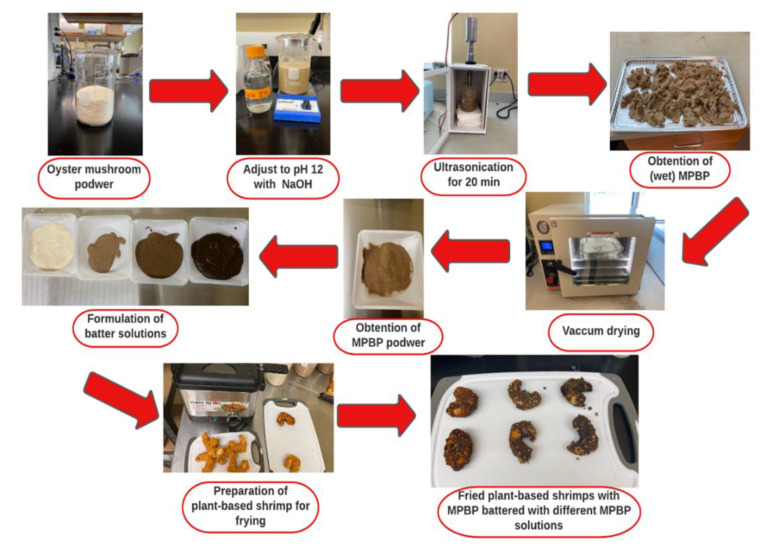
Processing flowchart to produce fried-batter coated shrimp coated with different batter formulations prepared with or without mushroom protein by-product.

**Figure 2 foods-12-00763-f002:**
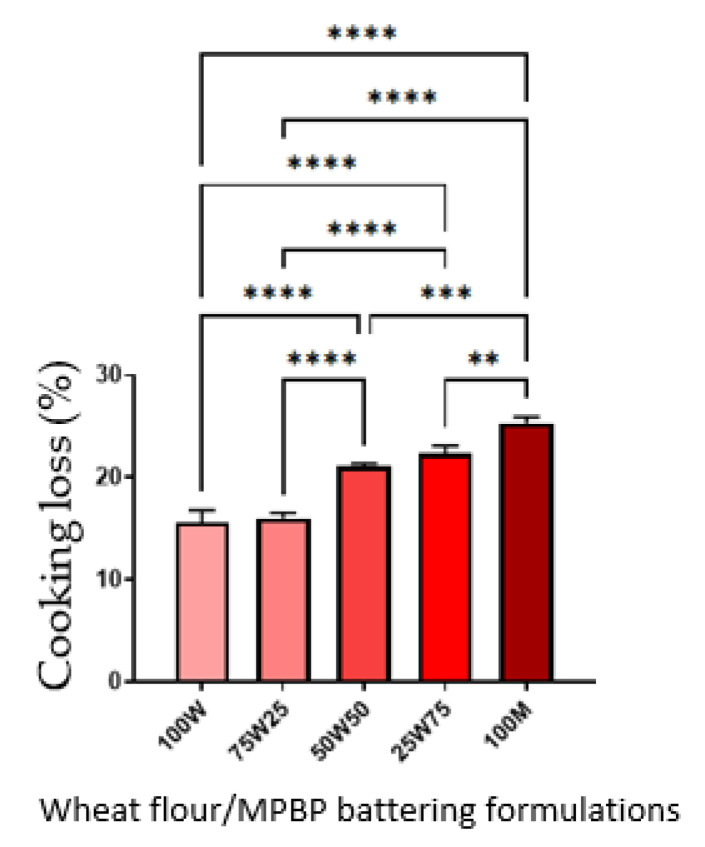
Cooking loss of batter-coated shrimp with MPBP. 100W: 100% wheat flour, 75W25: 75% wheat flour and 25% MPBP, 50W50: 50% wheat flour and 50% MPBP, 25W75: 25% wheat flour and 75% MPBP, 100M: 100. The error bars indicate standard division. ** *p* < 0.005, *** *p* < 0.0005, **** *p* < 0.00005.

**Figure 3 foods-12-00763-f003:**
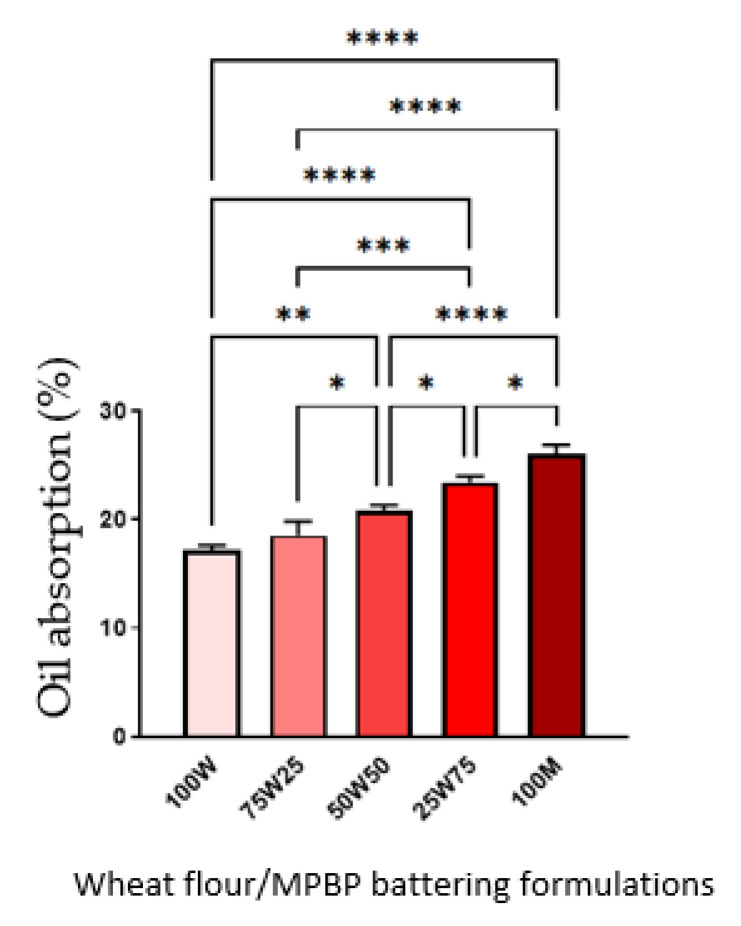
Oil absorption of batter-coated shrimp with MPBP. 100W: 100% wheat flour, 75W25: 75% wheat flour and 25% MPBP, 50W50: 50% wheat flour and 50% MPBP, 25W75: 25% wheat flour and 75% MPBP, 100M: 100. The error bars indicate standard division. * *p* < 0.05, ** *p* < 0.005, *** *p* < 0.0005, **** *p* < 0.00005.

**Figure 4 foods-12-00763-f004:**
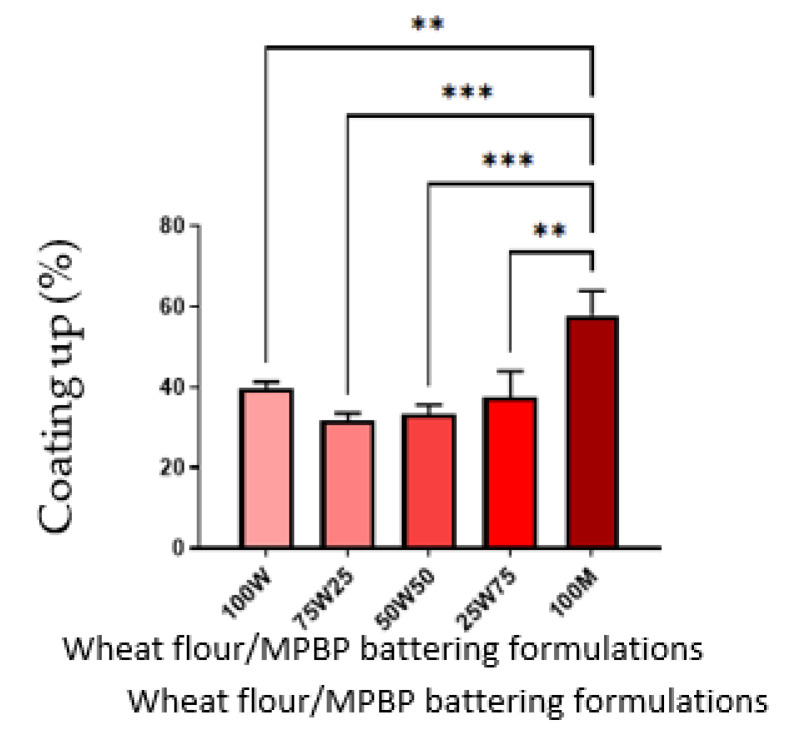
Coating of batter-coated shrimp with MPBP. 100 W: 100% wheat flour, 75 W25: 75% wheat flour and 25% MPBP, 50 W50: 50% wheat flour and 50% MPBP, 25 W75: 25% wheat flour and 75% MPBP, 100 M: 100. The error bars indicate standard division. ** *p* < 0.005, *** *p* < 0.0005.

**Figure 5 foods-12-00763-f005:**
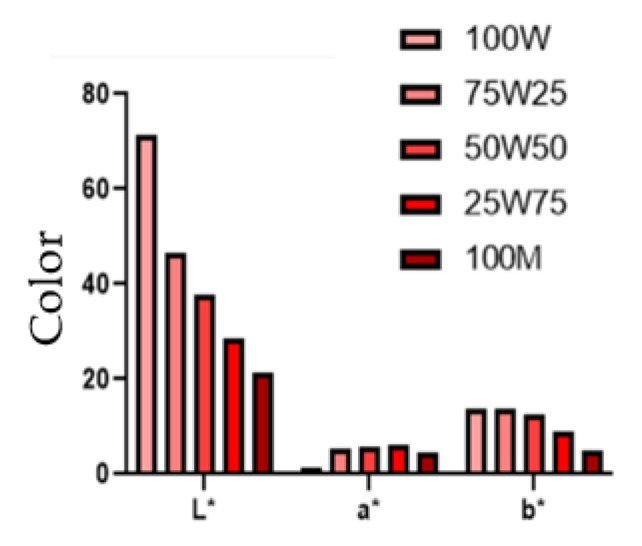
Color (*L**, *a**, *b*)* of batter-coated shrimp with MPBP. 100W: 100% wheat flour, 75W25: 75% wheat flour and 25% MPBP, 50W50: 50% wheat flour and 50% MPBP, 25W75: 25% wheat flour and 75% MPBP, 100M: 100.

**Table 1 foods-12-00763-t001:** Description of wheat flour/MPBP batter formulations.

Formulation	Wheat Flour	MPBP Powder
1	100%	0%
2	75%	25%
3	50%	50%
4	25%	75%
5	0%	100%

**Table 2 foods-12-00763-t002:** Chemical composition of the mushroom protein by-product.

Chemical Composition	
Moisture (%)	1.05 ± 0.83
Protein (%)	10.70 ± 0.70
Fat (%)	0.54 ± 0.03
Carbohydrates (%)	78.78 ± 0.89
Ash (%)	8.91 ± 0.12
Total Dietary Fiber (%)	56.6 ± 3.09
Insoluble Dietary Fiber (%)	49.3 ± 1.55
Soluble Dietary Fiber (%)	7.3 ± 0.34

Mean ± standard deviation is reported in triplicate measurements.

**Table 3 foods-12-00763-t003:** Physicochemical properties of the mushroom protein by-product.

Physicochemical Attributes	
pH	11.69 ± 0.05
Aw	0.044 ± 0.00
*L**	58.56 ± 2.45
*a**	5.61 ± 0.49
*b**	18.03 ± 1.23
35 mesh (500 µm) (%)	0.47 ± 0.08
60 mesh (250 µm) (%)	21.65 ± 2.76
120 mesh (125 µm) (%)	41.18 ± 3.89
230 mesh (63 µm) (%)	37.53 ± 2.67
<230 mesh (<63 µm) (%)	0.82 ± 0.07

**Table 4 foods-12-00763-t004:** Functional properties of mushroom protein by-products.

Functional Attributes	
Emulsifying Activity Index (m^2^/gr)	7.62 ± 0.07
Emulsion Stability Index (min)	52.42 ± 4.58
Water Holding Capacity (%)	4.88 ± 0.16
Solubility	4.87 ± 0.47
Oil Holding Capacity (%)	4.82 ± 0.15

Mean ± standard deviation is reported in triplicate measurements.

## Data Availability

No new data were created or analyzed in this study. Data sharing is not applicable to this article.

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
