# Peer review of "Total Utilization-Upcycling of Mushroom Protein By-Product: Characterization and Assessment as an Alternative Batter Ingredient for Fried Shrimp"

_foods, 2023, doi:10.3390/foods12040763_

Round 1

Reviewer 1 Report

The author should consider the below comments to improve the paper.

1.                    Why two abbreviations for the same term? As I understand from the abstract MPBP and M represent the same thing?! Do not define abbreviations multiply times. Please insert a list of abbreviations which would be useful considering the notations used in the paper.

2.                    How were the mushrooms dried? What method was used and in what conditions? What is the moisture content of the dried oyster mushrooms?

3.                    Line 146, after equations 2-3, the authors affirm that "Sw is the weight of the sample (g), ". There is no Sw in the equations??! What is the unit of DT (line 147)?

4.                    Line 207: please specify what type of oil was used.

5.                    Please use notations in equations 6-8 not words.

6.                    At line 230 please us m1 in accordance with equation 8.

7.                    Emulsion Stability Index (min) should be recalculated. There is a deviation of more than 50% in Table 4 which is not acceptable.

8.                   Please add x and y axis titles to all figures.

9.                   Conclusion section is too short and it needs to be consolidated with numerical findings and future perspectives should be added as well.

Author Response

  1. Why two abbreviations for the same term? As I understand from the abstract MPBP and M represent the same thing?! Do not define abbreviations multiply times. Please insert a list of abbreviations which would be useful considering the notations used in the paper.

Thank you for this comment, M was deleted and only MPBP was used. A list of Abbreviations is provided.

  1. How were the mushrooms dried? What method was used and in what conditions? What is the moisture content of the dried oyster mushrooms?

The mushrooms were purchased from Amazon, and the brand was Mushroom House (Westbury, NY). The moisture content was 10.57%.

  1. Line 146, after equations 2-3, the authors affirm that "Sw is the weight of the sample (g), ". There is no Sw in the equations?

Thank for this correction. Its well appreciated. Sw is the weight of the sample (g), was removed since it was not used to calculated  emulsifying activity index (EAI) and emulsion stability index (ESI). These properties for measure by spectrophotometry.

?! What is the unit of DT (line 147)?

Thank for this question that help clarification of DT, meaning the time differential (10 min) during absorbance recording.

  1. Line 207: please specify what type of oil was used.

Thank you for this comment, type of oil used was specified.

  1. Please use notations in equations 6-8 not words.

Thank for this suggestion, words were replaced by notations.

  1. At line 230 please us m1 in accordance with equation 8.

Thank you for this observation, m1 and m2 were replace for F1 and F2 in equation 8 to avoid confusion.

  1. Emulsion Stability Index (min) should be recalculated. There is a deviation of more than 50% in Table 4 which is not acceptable.

I apologize for this mistake. Instead of 24.48 it was 4.48. There was a typing mistake when writing the results in the table. Its well appreciated this observation.

  1. Please add x and y axis titles to all figures.

X and Y axis titles were added to all figures.

  1. Conclusion section is too short and it needs to be consolidated with numerical findings and future perspectives should be added as well.

Conclusion was improved as suggested.

Please feel free to check the manuscript version "Diego manuscript".

Reviewer 2 Report

Reviewer Comments

1. What do you mean by “plant-based shrimp”. This term is very confusing and might be revised as “batter coated shrimp” throughout the manuscript.

2. Line 40-42: Sentence is confusing kindly reframe.

3. Line 90-106: Kindly mention the reference method followed in the study for the separation of mushroom protein by-product.

4. Equation 2 & 3: Correct A_0 as A0

5. Line 159-161: Improve the description statement of Wtw, Wto and Wti as it seems little confusing.

6. Line 170-174: Why distinct flour: water ratio in different concentration of batter preparation is used as it can lead to the experimental biasness. The batter should be prepared using same water to flour ratio.

7. Table 1: There is no use of mentioning treatment column instead of it can be written as formulation 1,2,3,4, and 5

8. Line 263-265: How could natural ph of MPBP could be 11.69, when you have already adjusted the pH using 2M NAOH. Correct the statement.

9. Table 2: What is the reason of high (10%) residual protein in mushroom by-product?

10. Line 318-321: Oil uptake is directly proportional to the amount of moisture loss during frying. You have used different water to flour ratio during batter preparation which is leading to the difference in oil uptake in different formulation. This section needs more justification.

Author Response

1. What do you mean by “plant-based shrimp”. This term is very confusing and might be revised as “batter coated shrimp” throughout the manuscript.

Thank you for this suggestion, the term “plant-based shrimp” was replaced by “batter coated shrimp” throughout the manuscript

2. Line 40-42: Sentence is confusing kindly reframe.

Thank you for this comment,

The sentence:

“Recently, an increased interest has been shown in inedible mushrooms as a food ingredient for technological and functional applications, especially in high protein and clean-label markets.”

Has been rephrase to

“Recently, an increased interest has been shown in inedible mushrooms as a food ingredient for technological and functional applications, especially in high protein and clean-label markets.”

3. Line 90-106: Kindly mention the reference method followed in the study for the separation of mushroom protein by-product.

Thank you for this comment, the reference was mentioned.

Cruz-Solorio A., Villanueva-Arce, R., Garin-Aguilar ME., Leal-Lara H., Valencia-del Toro, G.         (2018). FUnctinoal properties of flours and protein concentrates of 3 strains of the edible           mushroom PLuerots ostreatus.

4. Equation 2 & 3: Correct A_0 as A0

Thank you for this correction, A_0 was change to A0 (0 as a subscript).

5. Line 159-161: Improve the description statement of Wtw, Wto and Wti as it seems little confusing.

the description statement of Wtw, Wto and Wti was improved.

6. Line 170-174: Why distinct flour: water ratio in different concentration of batter preparation is used as it can lead to the experimental biasness. The batter should be prepared using same water to flour ratio.

Thank you for this observation, the batter was prepared according to Rahimi & Ngadi et al., (2014) methodology. At the recommended water/ flour ratio, batter coated products acceptable fried products.

Rahimi, J.; Ngadi, M.O. Effect of batter formulation and pre-drying time on oil distribution fractions in fried batter. LWT 2014, 59, 820-826.

7. Table 1: There is no use of mentioning treatment column instead of it can be written as formulation 1,2,3,4, and 5

Thank you for this suggestion, treatment column was changed to formulation 1,2,3,4, and 5 in Table 1.

 8. Line 263-265: How could natural ph of MPBP could be 11.69, when you have already adjusted the pH using 2M NAOH. Correct the statement.

Thank you for this correction, the statement was corrected.

9. Table 2: What is the reason of high (10%) residual protein in mushroom by-product?

Thank you for this awareness, apparently Oyster Mushroom Cultivation Residue has reported similar values in a recent study (Moshtadhian et al., 2022).

Moshtaghian, H.; Parchami, M.; Rousta, K.; Lennartsson, P.R. Application of Oyster Mushroom Cultivation Residue as an Upcycled Ingredient for Developing Bread. Appl. Sci. 2022, 12, 11067. https://doi.org/10.3390/app122111067

10. Line 318-321: Oil uptake is directly proportional to the amount of moisture loss during frying. You have used different water to flour ratio during batter preparation which is leading to the difference in oil uptake in different formulation. This section needs more justification.

Thank you for this comment, the justification and explanation was improved of why Oil uptake is directly proportional to the amount of moisture loss during frying.

Please feel free to revise the manuscript version "Diego manuscript".
